# Semi-parametric model for timing of first childbirth after HIV diagnosis among women of childbearing age in Ibadan, Nigeria

**Joshua Odunayo Akinyemi**[1,2], **Rotimi Felix Afolabi**[1,3]*, **Olutosin Alaba Awolude**[4]

**1** Department of Epidemiology and Medical Statistics, College of Medicine, University of Ibadan, Ibadan, Nigeria, **2** Demography and Population Studies, Schools of Public Health and Social Sciences, University of the Witwatersrand, Johannesburg, South Africa, **3** Population and Health Research Entity, Faculty of Humanities, North-West University, Vanderbijlpark, South Africa, **4** Department of Obstetrics and Gynaecology, College of Medicine, University of Ibadan, Ibadan, Nigeria

* rotimifelix@yahoo.com

## Abstract

### Background

HIV diagnosis is a watershed in women's childbearing experience. It is usually accompanied by the fear of death and stigmatisation. Women diagnosed of HIV are often sceptical about pregnancy. Meanwhile, availability of antiretroviral treatments has impacted positively on childbearing experience among women living with HIV. We therefore investigated the timing of first childbirth after HIV diagnosis and its determinants among women in Ibadan, Nigeria.

### Methods

We extracted and analysed data from a 2015 cross-sectional study on childbearing progression among 933 women living with HIV and receiving care at University College Hospital, Ibadan, Nigeria. Extended Cox proportional hazards regression, a semi-parametric event history model was used at 5% significance level.

### Results

The women's mean age was 38.1 (± SD = 6.1) years and the median time to first birth after HIV diagnosis (FBI_HIV) was 8 years. The likelihood of first birth after HIV diagnosis was lower among women who desired more children (HR = 0.63, CI: 0.51–0.78). Women whose partners had primary and secondary education respectively were about 2.3 times more likely to shorten FBI_HIV compared to those whose partners had no formal education. Knowledge of partner's HIV-positive status (HR = 1.42, CI: 1.04,1.93) increased the likelihood of having a first birth after HIV diagnosis. Older age, longer duration on ART and a higher number of children at diagnosis were associated with a declined hazard of first birth after HIV diagnosis.

**Data Availability Statement:** Data cannot be shared publicly because of ethics policy at University of Ibadan. Data are available from the UI/ UCH Institutional Data Access / Ethics Committee

(contact via uiuchec@gmail.com) for researchers who meet the criteria for access to confidential data

**Funding:** The author(s) received no specific funding for this work.

**Competing interests:** The authors have declared that no competing interests exist.

## Conclusions

The median time to first childbirth after HIV diagnosis was long. Partner's HIV-positive status and higher educational attainment were associated with early childbearing after HIV diagnosis.

## Introduction

HIV prevalence remains a public health challenge in Nigeria. As the second epicentre of HIV infection globally, Nigeria contributed about 75% of new HIV infections in West and Central Africa in 2016 [1]. According to the recent UNAIDS report, the national HIV prevalence among adult aged 15–49 years is 1.5% [2], of which women have considerable higher prevalence of 1.9% compared to men [3]. Many of those individuals living with HIV are women who desire to affirm their motherhood. Nearly 50% and 75% of the women respectively desired more children according to studies conducted in Ethiopia and United Kingdom [4, 5].

Besides, as suggested by a study conducted in Malawi, women living with HIV (WLWH) desired to have their own children as a way of affirming their capacity for childbearing [6]. Meanwhile, a long duration on ART has been established to influence women's desire for children [7]. Even in the era of effective ART, empirical studies have established that HIV diagnosis may significantly influence women's decision and timing to have the next child birth or otherwise [8–11]. Child's birth may boost motherhood status among WLWH. Meanwhile, birth intervals could impact on the health of the mother and new-born in that sufficient spacing may avert adverse maternal and child outcomes among WLWH [12–14]. Therefore, knowledge of the time intervals to first childbirth after HIV diagnosis regarded as the first birth interval after HIV diagnosis (FBI_HIV) and its determinants are crucial to reproductive health care programming.

Previous studies have examined incidence and/or determinants of fertility experience after HIV diagnosis. For instance, studies conducted in Canada and Spain respectively have reported that about one-quarter and two-fifth of women were pregnant after HIV diagnosis [8, 9]. In Uganda, a sub-Saharan Africa country, about one-third of WLWH were pregnant within three years of ART initiation [11]. Socio-economic and demographic factors that could influence pregnancy or birth after HIV diagnosis include age at diagnosis [8, 15], duration since HIV diagnosis [6, 8], marital status [16], working status and parity [17]. With a total fertility rate of 5.3 births per woman [18] in Nigeria, and barely 50% ART access among people living with HIV, information on FBI_HIV is rare.

In the light of this, there is a need to explore the interval between HIV diagnosis and succeeding childbirth in a high fertility setting like Nigeria. Knowledge of FBI_HIV and its determinants can serve as evidence-based information for maternal and child health programmes and policy decision. The main goal of this study is therefore to assess the timing of first birth after HIV diagnosis among women of childbearing age in Ibadan, Nigeria and identify its associated factors. To address these objectives, a semi-parametric model for event history analysis was employed. This is necessary for objective and unbiased inferences. By and large, this would provide health officials with a better understanding of the attitudes of these women regarding the impact of HIV on childbearing and the factors associated with their reproductive decision-making.

## Methods

### Study design and setting

A cross-sectional survey on childbearing progression and proximate determinants of fertility among WLWH in Ibadan was conducted between November and December 2015. Data was

collected on full birth history for each woman which include birth order, sex, birthdate, time of birth (before or after HIV diagnosis), survival status (dead or alive), current age if alive, age at death (if dead) and preceding birth interval.

The study was conducted at the ART clinic of the University College Hospital (UCH). After information about the study was provided to all clinic attendees, nurses/counsellors referred the eligible consenting women to participate after every routine patients' education session. Over the two months period of data collection, an average of 25 participants was required on each clinic day. At every clinic visit, simple random sampling technique was used to select eligible consenting women. Sampling frame was the daily attendance register kept at the Record section of the clinic. Secret ballot-papers labelled "Yes" or "No" were prepared for the eligible women who registered on a clinic day. Of the total ballots, there were only 25 "Yes". Anyone who selected a "Yes" was enrolled to participate in the study after written informed consent was obtained.

Data were collected using pre-tested structured questionnaire. The interviewers who were female research assistants and postgraduate students in the College of Medicine were trained at a one-day workshop. During the training, they received general orientation about the study objectives, interviewing skills and health research ethics. Each question item was explained as well as how to record the responses. Data collection instrument (questionnaire) contained sections on socio-demographic characteristics, reproductive history, contraceptive knowledge and use, marital relationship and HIV care/treatment.

## Study population and variables

The study sample consisted of 933 consented women aged 18–49 years who had at least a childbirth before or after HIV diagnosis, enrolled for treatment (on ART) and care (not on ART) and supplied all required dates for relevant events for at least 12 months as at the time of the survey. All critically ill clients who were not well enough to provide responses were excluded.

**Outcome variable.**   The main outcome of interest in this study was FBI_HIV. It was estimated as number of years elapsed between date of HIV diagnosis and date of first birth thereafter. For proper event history analysis, women who did not have a childbirth after HIV diagnosis were right-censored and their time to event estimated as interval between date of HIV diagnosis and date of data collection.

**Explanatory variables.**   The explanatory variables considered for this study were women's socio-demographic characteristics (age at diagnosis, ethnicity, religion, education, employment and number of children at HIV diagnosis), marital profile (marital status at diagnosis, partner's education, partner's employment, desire more children and family setting) and HIV care profile (ART duration, status disclosure to partner and partner's HIV status). Age at diagnosis was categorised into <25, 25–34 and ≥35 years; number of children at diagnosis: 0, 1–2 and ≥3; ART duration: <4, 4–7 and ≥8 years; marital status at diagnosis: cohabiting before diagnosis and cohabiting after diagnosis.

## Data analysis

Survival analysis methods were employed to estimate the rate and determinants of FBI_HIV. Number of years elapsed between date of HIV diagnosis and first birth thereafter was the "failure time" for women who had had first birth after HIV diagnosis. Kaplan-Meier survival method and log-rank test respectively was used to describe women's FBI_HIV and examine its association with each of the covariates. Semi-parametric extended Cox proportional hazard regression model was thereafter applied to identify determinants of FBI_HIV.

**Kaplan-Meier survival method.** It is a nonparametric method that estimates survival function or probability of surviving beyond a given time $t$. Suppose $T$ is a random variable denoting survival time of a woman having her first childbirth after HIV diagnosis beyond a specific time $t$, then the probability of $T$ is the survival function $\{S(t)\}$ expressed as a function of a cumulative function $\{F(t)\}$:

$$S(t) = \int_t^\infty f(y)dy = P(T > t) = 1 - F(t) \tag{1}$$

$$F(t) = \int_0^t f(y)dy = P(T < t) \tag{2}$$

The F(t) is the cumulative probability that a woman has her first childbirth after HIV diagnosis before time $t$. Meanwhile, f(t) is the probability density function of the survival time $T$, defined as the probability that a woman has her first childbirth after HIV diagnosis per unit time in a short interval expressed as:

$$f(t) = \lim_{\triangle t \to 0} \left\{ \frac{P(t \leq T < t + \triangle t)}{\triangle t} \right\} \tag{3}$$

Equivalently, S(t) could be expressed as a hazard function $\{H(t)\}$:

$$\log[S(t)] = -H(t) \Rightarrow S(t) = e^{-H(t)} \tag{4}$$

Such that the conditional probability of experiencing first childbirth after HIV diagnosis within a short time interval (t, t + Δt) having survived till time $t$ could be expressed as:

$$h(t) = \lim_{\triangle t \to 0} \left\{ \frac{P(t \leq T \leq t + \triangle t | T > t)}{\triangle t} \right\} \tag{5}$$

If $n_i$ is the number of women who were exposed to the risk of having first birth after HIV diagnosis, censored women inclusive, before $i^{th}$ survival time $(t_i)$ and $l_i$ is the number of women who had first birth after HIV diagnosis at $t_i$, then Eq (6) below estimates the survival functions.

$$s(t) = \prod_{i=1}^{m} \left\{ \frac{n_i - l_i}{n_i} \right\} \ni t_m < t < t_{m+1}; \; s(t) = 1 \; if \; t < t_1 \tag{6}$$

where $m$ is the number of different failure times (i.e., experiencing first birth after HIV diagnosis)

The Kaplan-Meier estimate of the survivor function above gives a descriptive summary of FBI_HIV including the median survival time.

**Cox proportional hazards model.** The semi-parametric Cox proportional hazard regression model is expressed as:

$$h(t_t) = h_0(t)e^{b_1 x_1 i + b_2 x_2 i + \cdots + b_p x_p i} \tag{7}$$

and

$$\log \left\{ \frac{h(t_i)}{h_0(t)} \right\} = b_1 x_1 i + b_2 x_2 i + \cdots + b_p x_p i \tag{8}$$

where $b_j$ is the $j^{th}$ coefficient of the predictor variable $X_j$, $p$ is the number of independent variables, while $h_0(t)$ is the baseline hazard function.

An important assumption for the Cox regression model is the proportional hazard assumption which requires the hazard ratio (HR) to be constant over time. This was investigated

graphically; parallel curves indicate proportionality. Also, Schoenfeld residuals test was conducted in which $p < 0.05$ implies the violation of the proportional hazard assumption. Even though using the two approaches simultaneously is recommended, Schoenfeld residual test is more objective than the log-log survival plot [19, 20].

Commonly used non-proportional hazard models are stratified and extended Cox regression models [19, 21]. Even though stratified cox model is equally effective, effect of the variable used to stratify would not be obtained. Hence, extended Cox regression which suggests non-proportional hazards over time was employed in this study. It models time-dependent variable (s) interaction with time. The model is expressed as:

$$h(t_i, x(t)) = h_0(t)exp\{(b_1 x_{1i} + \cdots + b_{p1} x_{p1i}) + (a_1 x_{1i} g_1(t) + \cdots + a_{p2} x_{p2i} g_{p2}(t))\} \quad (9)$$

and

$$\log\left\{\frac{h(t_i, x(t))}{h_0(t)}\right\} = \left(b_1 x_{1i} + \cdots + b_{p1} x_{p1i}\right) + (a_1 x_{1i} g_1(t) + \cdots + a_{p2} x_{p2i} g_{p2}(t)) \quad (10)$$

where

$p = p_1 + p_2$ predictors such that $p_2$ predictor(s) interact with time since proportional assumption fails

$a_j = j^{th}$ overall effect of $X_j(t) = X_j \times g_j(t)$ (time dependent predictor(s)) such that its positive value indicates increase in hazard as time-to-event increases; otherwise, it decreases

$$g_j(t) = \begin{cases} 0, & PH\ assumption\ met \\ t, & interaction\ X_j t\ for\ jth\ variable \\ In(t), & interaction\ X_j In(t)\ for\ jth\ variable \\ heaviside\ function, & constant\ HR\ for\ different\ time\ intervals \end{cases}$$ - time function of

variable.

The coefficient $b_j$ indicates the changes in the expected duration of FBI_HIV for every unit change in the $j^{th}$ predictor. The exponentials of the coefficients suggest the tendency of a woman exposure to having a first childbirth after HIV diagnosis; thus, HR >1 indicates higher exposure, HR < 1 lower exposure and HR = 1 equally likely exposure. All significant variables ($p \leq 0.25$) emanated from the log-rank test including clinically important variable(s) were included in the extended Cox regression model [22]. All analyses were conducted at 5% level of significance using STATA 14 SE (StataCorp LP, College Station, USA).

## Ethical approval

The University of Ibadan/University College Hospital Instituional Review committee approved the survey protocol with approval number (UI/EC/15/0230). Participants gave informed consent and were informed of their freedom to withdraw from interview at any point, prior to data collection. Every tenet of Helsinki declaration and other ethical requirements were strictly complied with throughout the study. No identifying information was collected from participants and study questionnaires were accessible to only investigators and authorised research staff.

## Results

### Women's characteristics

Table 1 presents the women descriptive statistics, median survival time and log-rank test of survival curves equality. The mean age of the women was 38.1(±SD = 6.1) years (though not

**Table 1. Characteristics of women aged 18–49 years diagnosed of HIV in Ibadan, Nigeria.**

| Characteristic | Total at risk | | Had first birth (n = 413) | MF | p-value^ |
|---|---|---|---|---|---|
| | n | % | % | years | |
| **Age at diagnosis (years)** | | | | | <0.001*** |
| <25 | 114 | 12.2 | 67.5 | 4 | |
| 25–34 | 503 | 53.9 | 55.9 | 5 | |
| ≥35 | 316 | 33.9 | 17.4 | na | |
| **Education** | | | | | 0.001*** |
| None | 63 | 6.8 | 25.4 | na | |
| Primary | 195 | 20.9 | 37.9 | 14 | |
| Secondary | 443 | 47.5 | 44.7 | 7 | |
| Higher | 232 | 24.9 | 53.9 | 5 | |
| **Employment** | | | | | 0.027** |
| Not working | 90 | 9.6 | 54.4 | 4 | |
| Working | 843 | 90.4 | 43.2 | 8 | |
| **Ethnicity** | | | | | 0.001*** |
| Non-Yoruba | 182 | 19.5 | 53.3 | 5 | |
| Yoruba | 751 | 80.5 | 42.1 | 9 | |
| **Religion** | | | | | 0.348 |
| Christian | 567 | 60.8 | 45.9 | 7 | |
| Islam | 366 | 39.2 | 41.8 | 9 | |
| **No of children at diagnosis** | | | | | <0.001*** |
| None | 207 | 22.2 | 73.4 | 3 | |
| 1–2 | 444 | 47.6 | 46.2 | 6 | |
| ≥3 | 282 | 30.2 | 19.9 | 14 | |
| **Marital status at diagnosis[+]** | | | | | <0.001*** |
| Cohabiting after diagnosis | 105 | 11.4 | 74.3 | 4 | |
| Cohabiting before diagnosis | 819 | 88.6 | 40.4 | 10 | |
| **Partner education[+]** | | | | | 0.021** |
| None | 38 | 4.1 | 21.1 | na | |
| Primary | 99 | 10.6 | 39.4 | na | |
| Secondary | 444 | 47.6 | 43.9 | 7 | |
| Higher | 337 | 36.1 | 49.6 | 6 | |
| **Partner employment** | | | | | <0.001*** |
| Working | 794 | 85.1 | 47.0 | 7 | |
| Not working | 139 | 14.9 | 28.8 | na | |
| **Desire for more children** | | | | | 0.258* |
| No more | 433 | 46.4 | 42.0 | 9 | |
| Desired | 500 | 53.6 | 46.2 | 7 | |
| **Family setting[+]** | | | | | <0.001*** |
| Monogamy | 596 | 65.6 | 51.0 | 6 | |
| Polygamy | 312 | 34.4 | 32.1 | na | |
| **ART use duration (years)** | | | | | 0.169* |
| <4 | 324 | 34.7 | 36.4 | 5 | |
| 4–7 | 347 | 37.2 | 45.8 | 8 | |
| ≥8 | 262 | 28.1 | 51.9 | 9 | |
| **Status disclosure to partner** | | | | | 0.024** |
| Non-disclosure | 124 | 13.3 | 50.8 | 5 | |
| Disclosure | 809 | 86.7 | 43.3 | 9 | |

(*Continued*)

**Table 1.** (Continued)

| Characteristic | Total at risk | | Had first birth (n = 413) | MF | p-value^ |
|---|---|---|---|---|---|
| | n | % | % | years | |
| **Partner HIV status** | | | | | 0.002*** |
| Don't know | 228 | 24.4 | 32.0 | 17 | |
| Positive | 257 | 27.5 | 44.7 | 8 | |
| Negative | 448 | 48.0 | 50.2 | 6 | |
| *Total* | 933 | | 44.3 | 8 | |

* p≤.25

** p < .05

***p < .001

^ based on log-rank test

+ missing not reported; n–number of women; MF—Median FBI_HIV; na–MF cannot be computed owing to category's low percentage of women who had first birth after HIV diagnosis

displayed in Table 1). Women aged <25 (12.2%) and 25–34 years (53.9%) constituted the least and the highest proportion of women studied respectively. Most women had secondary education (47.5%) and desired more children (53.6%). The result also showed 37.2% of the women had been on ART for 4–7 years, while 48.0% had HIV-negative partners. Majority of the women were Yoruba (80.5%), employed (90.4%), had employed partner (85.1%), disclosed status to partner (86.7%), and had at least a living child at diagnosis (77.8%).

Of 933 women, 413 (44.3%) had had first birth after being diagnosed of HIV as at the survey date. Women who were cohabiting before being diagnosed (10 years) and had primary education (14 years), ≥3 children at diagnosis (14 years) or no knowledge of partner's HIV status (17 years) delayed the first birth in at least 10 years after HIV diagnosis. Nearly all the selected variables had significant (p≤0.25) differences in their respective category's survival curves except for religion. For instance, a significantly higher percentage of women aged <25 years (67.5%) and those who had been using ART for at least 8 years prior to the date of interview (51.9%), had no child before HIV diagnosis (73.4%) or cohabited prior to HIV diagnosis (74.3%) gave birth after HIV diagnosis. The median time to first birth after HIV diagnosis was 8 years (Table 1). This is also revealed in Fig 1, which demonstrates the probabilities of the risk of having a first birth after the HIV diagnosis and the survival rate.

## Investigation of proportional hazard assumption

Asides the Schoenfeld test, the graphical assessment of the proportional hazard assumption is shown in Fig 2. The outcome showed that proportional hazard assumption failed (global test: p = 0.015). Further investigation revealed that age at diagnosis (p = 0.014) violated the proportional hazard assumption. Extended cox regression was considered for further analysis as it suffices in handling non-proportional hazards.

## Predictors of time to first birth after HIV diagnosis

The outcomes of the extended Cox regression model are presented in Table 2. Age at diagnosis (time-varying variable), ART use duration, desire for more children, number of children at diagnosis, partner's education and knowledge of partner's HIV status were significantly associated with FBI_HIV. Women who desired for more children after HIV diagnosis (HR = 0.63; CI: 0.51–0.78; p<0.001) had a 37% reduced likelihood of shortening FBI_HIV compared with those who want no more. Increasing years of ART use was associated with a lower hazard of

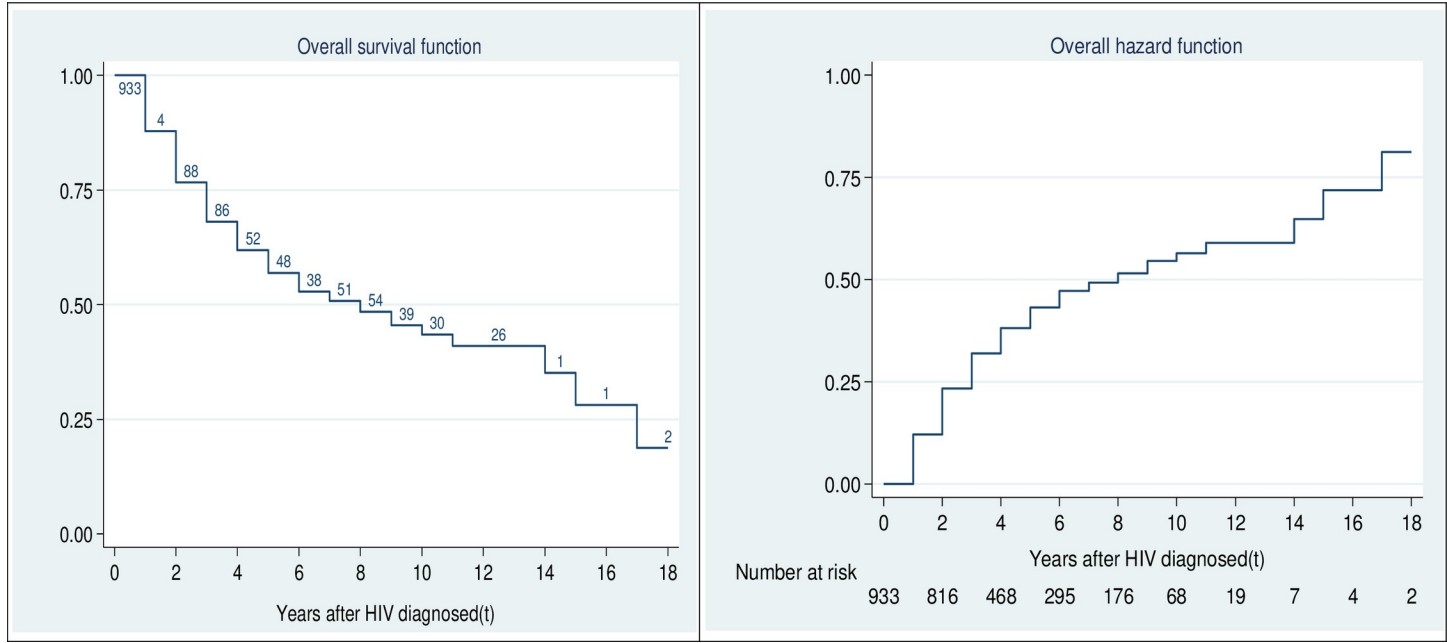

**Fig 1. Overall survival and hazard function of FBI_HIV.** The probability plot displaying the cumulative survival curve and the risk of having a first birth after HIV diagnosis.

shortened FBI_HIV. For instance, women who had used ART for 4–7 (HR = 0.74; CI: 0.59–0.92; p = 0.006) and ≥ 8 (aHR = 0.78; CI: 0.44–0.70; p = 0.034) years were more likely to delay first childbirth after HIV diagnosis compared with their counterpart with a short duration of ART use. The possibility of a first birth after being diagnosed of HIV declined as number of children at diagnosis increased. Women who had 1–2 (HR = 0.55; CI: 0.44–0.70; p<0.001) and >2 (HR = 0.24; CI: 0.17–0.35; p<0.001) children at diagnosis respectively had 45% and 76% reduced risks of having first birth relative to women with no surviving child at diagnosis. Interestingly, women whose partners had primary and secondary education respectively were about 2.3 times more likely of having first birth after HIV diagnosis compared to those whose partners had no formal education. Similarly, women who had the knowledge of their partners' HIV-positive status (HR = 1.42; CI: 1.04–1.93; p<0.027) were about 42% more likely to have first birth after HIV diagnosis relative to those who did not know their partners' HIV status. Although older age was insignificantly associated with higher likelihood of having a first birth at the time of diagnosis, older age lengthened FBI_HIV subsequently. Such that the hazard of having a first birth after HIV diagnosis was about 35% less likely among women aged ≥35 years (HR = 0.64, CI: 0.51–0.80; p<0.001) compared to those aged <25 years.

## Discussion

This study was conducted to examine the timing of first birth after HIV diagnosis among women of reproductive age who attend ART clinic in Ibadan Nigeria. Despite most women having at least one child at the time of HIV diagnosis, nearly half had first birth after HIV diagnosis. This result buttresses the importance of motherhood as a means of coping with HIV diagnosis [10], and the need to integrate reproductive counselling into HIV treatments and care in Nigeria. Although the available literature reported one-quarter of women being pregnant after HIV diagnosis in Canada [9], the percentage of women having at least a child after

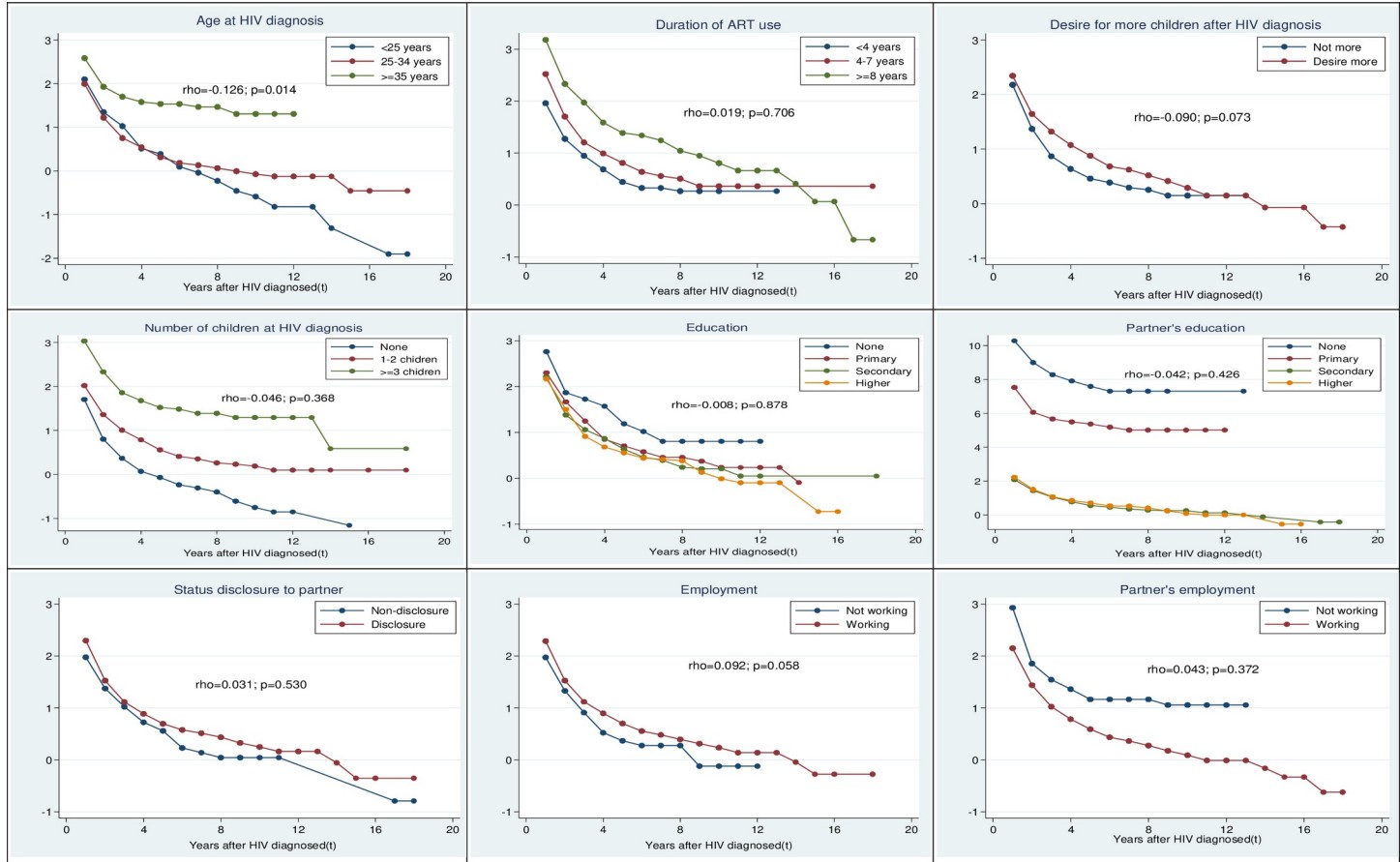

**Fig 2. Log-log survival plot.** This plot shows a graphical examination of the proportional hazard assumption.

HIV diagnosis is high in Nigeria. By implication, it is also higher than 39% found in a cross-sectional study conducted among women aged 18–49 years in Spain [8].

Of note, our finding showed that the median first birth interval after HIV diagnosis (8 years) was long. This interval is higher compared to the findings of studies conducted in Brazil, Spain and Uganda which reported 2, ≤ 2 and 3.8 years interval between HIV diagnosis and the first pregnancy, respectively [8, 11, 23]. The birth interval is a bit longer among the women studied compared to the median birth interval of about 2.6 years among the general population of women of reproductive age in Nigeria [24]. This study revealed that FBI_HIV in Nigeria is influenced by age at diagnosis, ART use duration, desire for more children, increased in number of children at diagnosis, and having educated partner or knowledge of partner's HIV status.

Interestingly, despite the insignificant time-varying effect of age at diagnosis, the likelihood of having a first birth after HIV diagnosis significantly decreased with age. This result aligns with an earlier study [11] conducted in Uganda among HIV-positive women of childbearing age which concluded that being younger is significantly associated with pregnancy risk. Of course, women are more likely to delay first childbirth till their advanced reproductive age as observed in this study; this is consistent with the finding of a study in Colombia [15]. This may suggest an urgent sensitisation and advocacy on the related pros and cons of postponing childbirth till older childbearing age among WLHW.

**Table 2. Effect of selected characteristics on time to first birth after HIV diagnosis among women aged 18–49 years in Ibadan, Nigeria.**

| Characteristics (n = 903) | HR | 95% CI | p-value |
|---|---|---|---|
| **Age at diagnosis** | | | |
| <25 | Ref | | |
| 25–34 | 1.32 | 0.83,2.10 | 0.240 |
| > = 35 | 1.14 | 0.57,2.27 | 0.715 |
| **Educational level** | | | |
| None | Ref | | |
| Primary | 1.42 | 0.81,2.51 | 0.225 |
| Secondary | 1.58 | 0.91,2.74 | 0.104 |
| Higher | 1.63 | 0.91,2.92 | 0.099 |
| **Employment Status** | | | |
| Not working | Ref | | |
| Working | 0.88 | 0.64,1.21 | 0.433 |
| **Ethnicity** | | | |
| Non-Yoruba | Ref | | |
| Yoruba | 0.84 | 0.66,1.07 | 0.155 |
| **No of children at diagnosis (Q505_b) Q505N** | | | |
| None | Ref | | |
| 1–2 | 0.55 | 0.44,0.70 | 0.000*** |
| > = 3 | 0.24 | 0.17,0.35 | 0.000*** |
| **Marital status at diagnosis (marital_b)** | | | |
| Cohabiting before diagnosis | Ref | | |
| Cohabiting after diagnosis | 0.97 | 0.71,1.33 | 0.853 |
| **Partner education** | | | |
| None | Ref | | |
| Primary | 2.35 | 1.08,5.13 | 0.031* |
| Secondary | 2.30 | 1.11,4.79 | 0.026* |
| Higher | 1.87 | 0.89,3.94 | 0.098 |
| **Partner employment (Q407_b) (Q107)** | | | |
| Not working | Ref | | |
| Working | 1.37 | 0.95,1.98 | 0.091 |
| **Family type** | | | |
| Monogamy | Ref | | |
| Polygamy | 0.83 | 0.65,1.06 | 0.132 |
| **Desire for more children** | | | |
| No more | Ref | | |
| desired | 0.63 | 0.51,0.78 | 0.000*** |
| **ART use duration** | | | |
| <4 | Ref | | |
| 4–7 | 0.74 | 0.59,0.92 | 0.006** |
| ≥8 | 0.57 | 0.40,0.81 | 0.002** |
| **Status disclosure to partner** | | | |
| Non-disclosed | Ref | | |
| Disclosed | 0.94 | 0.71,1.24 | 0.658 |
| **Partner's HIV status** | | | |
| Don't know | Ref | | |
| Positive | 1.42 | 1.04,1.93 | 0.027* |

(*Continued*)

**Table 2.** (Continued)

| Characteristics (n = 903) | HR | 95% CI | p-value |
|---|---|---|---|
| Negative | 1.22 | 0.92,1.63 | 0.173 |
| **Age at diagnosis*t** | | | |
| <25*time | Ref | | |
| 25–34*time | 0.88 | 0.79,0.98 | 0.020* |
| ≥35*time | 0.64 | 0.51,0.80 | 0.000*** |

* p < .05

** p < .01

***p < .001; HR–adjusted hazard ratio; CI–Confidence interval; Ref–reference category

While Kaida and colleagues [11] opined that disclosure of HIV status to partner is significantly associated with the risks of being pregnant after diagnosis, our study suggested that a knowledge of partner's HIV-positive status significantly shortened the interval. This may likely suggest a concluded agreement between the women and their respective partners to increasingly give birth early after HIV diagnosis, perhaps due to enrolment for ART treatment and care. This has an implication for the women childbearing planning considering the vital role of partners in fertility decision-makings. However, nearly three-quarter of women confirmed either partner's HIV-unknown or HIV-negative status as observed in this study; this is an indication of partners at-risk population for HIV infection. This therefore calls for an all-inclusive reproductive healthcare and conception counselling programmes for WLWH and their respective partners to be integrated into HIV care programmes [15].

It is also worth noting that the tendency of women to have first birth after HIV diagnosis was higher among those whose partners are educated. Contrary to the belief that higher education improves women's social and economic status and offers women access to non-childbearing activities, having educated partner stimulated childbirth after HIV diagnosis. Other empirical literature has established the role of partner decision-making in women reproductive process which could be linked to the knowledge that ART could prevent either partner or their children from being HIV-positive [25]. This, coupled with the idea of showcasing reproductive prowess, may motivate the considerable increase in number of childbirth after HIV diagnosis [6, 8, 17].

Furthermore, increasing years of ART use or desire for more children increases the FBI_HIV in consistent with similar studies in other settings [6, 7]. This is not surprising as the association between ART use duration and fertility desire has been reported in literature [7]. This may be hinged on the believe that the longer the duration on ART, the higher the improved quality of life and consequently the more likely to raise children free from HIV infection. Another possible reason may be associated with the tendency of the women to consolidate their relationship by having more children as most WLWH in relationships usually have partners who are yet to father a child [5, 6].

In addition, increased number of children at diagnosis had a protective effect against shortened FBI_HIV as women were less likely to have children either in the short or long term. This finding corroborates previous studies in other contexts [26–28]. As women attain their desired family size, the likelihood to desire or have additional child may decrease or cease altogether [29].

## Limitations of the study

This study has its limitations. With a cross-sectional design, in-depth analysis of temporal relationship between HIV diagnosis and childbearing transitions could not be carried out. Besides,

self-reported data collected retrospectively with no means of verification may influence recall bias. This, however, has been overcome by the usage of carefully designed questionnaire and trained interviewers to reduce perceived bias to the barest minimum.

## Conclusions

In conclusion, information on fertility timing after HIV diagnosis is necessary for care and management of people living with HIV. The percentage of women who had first birth after HIV diagnosis is considerable with a high median time to first birth after HV diagnosis. Several factors including advanced women age at diagnosis, duration of ART use, desire for more children and number of children at HIV diagnosis have been identified as risk factors of women's first birth interval after HIV diagnosis. Other factors include having partners who were HIV-positive or attained formal education. These identified factors should be integrated into HIV care program design and implementation.

## Acknowledgments

We appreciate the cooperation of study participants and staff members at the UCH Antiretroviral Clinic, Ibadan.

## Author Contributions

**Conceptualization:** Joshua Odunayo Akinyemi, Rotimi Felix Afolabi.

**Formal analysis:** Joshua Odunayo Akinyemi, Rotimi Felix Afolabi.

**Writing – original draft:** Joshua Odunayo Akinyemi, Rotimi Felix Afolabi, Olutosin Alaba Awolude.

**Writing – review & editing:** Joshua Odunayo Akinyemi, Rotimi Felix Afolabi, Olutosin Alaba Awolude.

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
