## [Decision Letter · Decision Letter 0]

2 Jul 2020

PONE-D-20-04600

Semi-parametric model for time to first birth after HIV diagnosis among women in South-West, Nigeria

PLOS ONE

Dear Dr. Afolabi,

Thank you for submitting your manuscript to PLOS ONE. After careful consideration, we feel that it has merit but does not fully meet PLOS ONE’s publication criteria as it currently stands. Therefore, we invite you to submit a revised version of the manuscript that addresses the points raised during the review process.

We look forward to receiving your revised manuscript.

Kind regards,

Kannan Navaneetham, PhD

Academic Editor

PLOS ONE

Journal Requirements:

'The project from which this data was extracted was supported

by the Medical Education Partnership Initiative in Nigeria (MEPIN) project funded by

Fogarty International Centre, Office of AIDS Research, and the National Human Genome

Research Institute of the National Institute of Health, the Health Resources and Services

Administration (HRSA) and the Office of the U.S. Global AIDS Coordinator under Award

Number R24TW008878.

'The author(s) received no specific funding for this work.'

Additional Editor Comments (if provided):

Reviewers' comments:

Reviewer's Responses to Questions

**Comments to the Author**

1. Is the manuscript technically sound, and do the data support the conclusions?

Reviewer #1: Yes

Reviewer #2: Partly

2. Has the statistical analysis been performed appropriately and rigorously? 

Reviewer #1: Yes

Reviewer #2: No

3. Have the authors made all data underlying the findings in their manuscript fully available?

Reviewer #1: No

Reviewer #2: No

4. Is the manuscript presented in an intelligible fashion and written in standard English?

Reviewer #1: Yes

Reviewer #2: No

5. Review Comments to the Author

Reviewer #1: Comments

1. The title should be modified as per the study objectives.

2. The abstract is too long (particularly, result part). Try to make it short and precise.

3. Some edition is needed. (eg: line 28, 32 pronoun “we” is wrongly used).

4. How much of the data were 'valid' and what was the level of missingness for the variables of interest that you used?

5. Be consistent in your references citation (eg. Line 74-line 78 different reference citation )

6. On Line 124: you have stated that “women aged 18-49 years who had had at least a child-birth…. ” were considered in the study. From line 129-30 it has been stated that “For proper event history analysis, women who did not have a child birth after HIV diagnosis were censored…. ” which contradicts the former statement. Take a look.

7. Line 211: why you need to have exponential distribution test?

8. Line 245: Employment status (p=0.049) violated the proportional hazard assumption but the graph for employment status (Fig 2) is parallel which implied that PH assumption is not violated. How could it be?

9. Table 2 on page 13: It has been identified that Cox PH model is not appropriate since PH assumption is violated for some explanatory variables. Once the violation of PH assumption is detected, no need to keep cox PH model for comparison purpose with extended cox model. It is better to consider only the extended model and interpret the result. Model comparison should be made (using AIC, BIC, ) only between appropriate models for the particular data.

10. Interpretation of HR is not clear.

11. Be consistent when you write the response variable (first birth interval after diagnosis or time to first birth after diagnosis). Line 337 “first birth after HIV diagnosis” is not good expression.

12. Inclusion exclusion criteria are not clearly stated.

13. Limitation should be separated from discussion.

Reviewer #2: Generally the authors need to check on grammatical mistakes that abound in the manuscript. Some have been highlighted.

Data Analysis

Kaplan Meier.

The notations and definitions are misleading. The authors use the random variable T to denote a woman who had her FBI_HIV beyond t time. Normally T denotes the future lifetime random variable of an individual aged 0(newborn) .In this context, T could be time to first birth after HIV diagnosis, so that s(t0 is the probability of birth after t years.

Equation 2 is wrong, the integral should be from 0 to t and not 1.

Following the author’s arguments, the probable definition of f(t) is the probability that a woman has a child at time t.

The equations are poorly written, kindly use latex, Microsoft equation etc.

The author is also silent on the censoring mechanism assumed in the data.

Line 160: Mixed up notations: What does t_i represent? Survival times, or failure times/birth times. There is also a general mix up under this heading.

COX PH

Line 190, 191: Not clear

Women Characteristics

Although the author states that table 1 displays among others the Kaplan Meier survival estimates, I have been unable to see the Kaplan Meier survival estimates.

Results and discussions

It would be of great value for the authors to outline how they selected the covariates. What feature selection techniques were employed?

Important features like use of ART and status of the partner have been left out.

6. PLOS authors have the option to publish the peer review history of their article (what does this mean?). If published, this will include your full peer review and any attached files.

Reviewer #1: **Yes: **Ayele Gebeyehu Chernet

Reviewer #2: **Yes: **Elphas Okango

---

## [Author Response · Author response to Decision Letter 0]

17 Jul 2020

The comments are valuable in terms of enriching the quality of the paper.

Thank you all.

---

## [Decision Letter · Decision Letter 1]

16 Sep 2020

PONE-D-20-04600R1

Semi-parametric model for timing of first childbirth after HIV diagnosis among women of childbearing age in Ibadan, Nigeria

PLOS ONE

Dear Dr. Afolabi,

Thank you for submitting your manuscript to PLOS ONE. After careful consideration, we feel that it has merit but does not fully meet PLOS ONE’s publication criteria as it currently stands. Therefore, we invite you to submit a revised version of the manuscript that addresses the points raised during the review process.

We look forward to receiving your revised manuscript.

Kind regards,

Kannan Navaneetham, PhD

Academic Editor

PLOS ONE

Reviewers' comments:

Reviewer's Responses to Questions

**Comments to the Author**

1. If the authors have adequately addressed your comments raised in a previous round of review and you feel that this manuscript is now acceptable for publication, you may indicate that here to bypass the “Comments to the Author” section, enter your conflict of interest statement in the “Confidential to Editor” section, and submit your "Accept" recommendation.

Reviewer #2: (No Response)

2. Is the manuscript technically sound, and do the data support the conclusions?

Reviewer #2: Yes

3. Has the statistical analysis been performed appropriately and rigorously? 

Reviewer #2: Yes

4. Have the authors made all data underlying the findings in their manuscript fully available?

Reviewer #2: No

5. Is the manuscript presented in an intelligible fashion and written in standard English?

Reviewer #2: No

6. Review Comments to the Author

Reviewer #2: The authors have made a great improvement on the manuscript, however there are still some issues:

There still exists some grammatical mistakes e.g line 25, 61, 213, 237 etc.

Line 85: Which data is the author talking about?

Line 170: use math type, Microsoft equation, latex, or r markdown to type equations/ mathematical symbols.

Line 192. The coefficients b_js are not used to indicate statistical significance, but the strength and direction of relationship.

7. PLOS authors have the option to publish the peer review history of their article (what does this mean?). If published, this will include your full peer review and any attached files.

Reviewer #2: No

---

## [Author Response · Author response to Decision Letter 1]

21 Sep 2020

Thank you for the recommendation.

---

## [Editor Report · Decision Letter 2]

23 Sep 2020

Semi-parametric model for timing of first childbirth after HIV diagnosis among women of childbearing age in Ibadan, Nigeria

PONE-D-20-04600R2

Dear Dr. Afolabi,

We’re pleased to inform you that your manuscript has been judged scientifically suitable for publication and will be formally accepted for publication once it meets all outstanding technical requirements.

Kind regards,

Kannan Navaneetham, PhD

Academic Editor

PLOS ONE
---

## [Editor Report · Acceptance letter]

28 Sep 2020

PONE-D-20-04600R2 

Semi-parametric model for timing of first childbirth after HIV diagnosis among women of childbearing age in Ibadan, Nigeria 

Dear Dr. Afolabi:

I'm pleased to inform you that your manuscript has been deemed suitable for publication in PLOS ONE. Congratulations! Your manuscript is now with our production department. 

Kind regards, 

on behalf of

Professor Kannan Navaneetham 

Academic Editor

PLOS ONE